# Sustained Phosphorus Removal by Calcareous Materials in Long-Term (Two Years) Column Experiment

**Solvei Mundbjerg Jensen** [1,2,3,*], **Helmer Søhoel** [4], **Frances Helen Blaikie** [4,5], **Carlos Alberto Arias** [1,2]
**and Hans Brix** [1,2]

1   Department of Biology, Aarhus University, Ole Worms Allé 1, Aarhus C, 8000 Aarhus, Denmark;
    carlos.arias@bio.au.dk (C.A.A.); hans.brix@bio.au.dk (H.B.)
2   Centre for Water Technology (WATEC), Aarhus University, Ny Munkegade 120, Aarhus C,
    8000 Aarhus, Denmark
3   Sino-Danish Centre for Education and Research (SDC), Niels Jensens Vej 2, Aarhus C, 8000 Aarhus, Denmark
4   Coating and Polymer Technology, Danish Technological Institute (DTI), Kongsvang Allé 29, Aarhus C,
    8000 Aarhus, Denmark; helmer.sohoel@gmail.com (H.S.); frances@blaikiesciencesolutions.co.nz (F.H.B.)
5   Blaikie Science Solutions, Invermay AgResearch Campus, Puddle Alley 176, Mosgiel 9053, New Zealand
*   Correspondence: solvei.mundbjerg@bio.au.dk

**Abstract:** (1) Phosphorus (P) removal has proven difficult in decentralized wastewater treatment systems, and external filters installed with a highly P sorbent material have been proposed to improve the P removal. In particular, calcium (Ca) rich materials have shown promising results. (2) Eight materials (five calcareous materials, one quartz sand, and two Sol–Gel coated calcareous materials) were tested in columns fed with P-spiked tap water for two years. The experiment was operated under four periods with increased P concentration from 3.3 to 21.5 mg P L$^{-1}$, and with increased surface loading rate from 18 to 227 mm d$^{-1}$. After termination, the element content was measured in four column height fractions. (3) Initially, all columns removed P effectively and the calcareous materials (CAT, CAT A, and CAT C) maintained an effective removal until termination, while increases in effluent P concentration were detected already after 7 weeks for SAN and after 80–90 weeks for OPO, PHO, CAL, and HYG. The highest P content for materials were measured for the bottom fraction closest to the inlet distribution. For most materials, we observed a good agreement between the maximum sorption capacity ($Q_{max}$) and the P content in the bottom fraction; however, a discrepancy was observed for CAL, CAT A, and CAT C. (4) In conclusion, the calcareous materials provided a consistent P removal for all 24 months. Additionally, the Sol–Gel coating had a minimal effect on the P removal capacity contrary to previous findings in batch experiments for the coated materials.

**Keywords:** phosphorus; filter; material; media; Sol–Gel coating; total P binding capacity; sorption; Langmuir; constructed wetlands; treatment wetlands

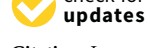



## 1. Introduction

Anthropogenic activities generate wastewater that, if released untreated, cause harmful consequences in receiving waters, e.g., eutrophication and deterioration of aquatic environments due to excessive nutrients. However, centralized wastewater treatment solutions are not always feasible to implement, and decentralized solutions have been developed targeting the affected waters closer to the source of pollution [1]. There is a large range of decentralized solutions, with one group being nature-based solutions (NBS), covering a wide span of effective solutions such as Treatment Wetlands (TWs, also known as Constructed Wetlands). TWs are human-made wetlands engineered to optimize physical and biochemical processes occurring in natural wetlands to reduce pollutants [2,3]. Lately, TWs have gained popularity as they are relatively simple to construct, easily maintained, and require low external energy input, making them an appropriate technology for decentralized wastewater treatment [4,5]. In addition, TWs are robust systems with

reliable and effective removal of pollutants and therefore ideal in rural areas for the treatment of wastewaters from dwellings and in developing countries and sites in need of onsite solutions.

Although most of the decentralized solutions are effective at mitigating the effect of polluted waters, meeting the P removal requirements is typically a challenge. For example, in Denmark only a few on-site systems, such as zero-discharge evaporative willow systems and soil infiltration systems, can fulfill the strictest P removal requirements of 90% P reduction or maximum outlet P concentration of 1.5 mg $L^{-1}$ [6,7]. The pollutant removal processes in TWs consist of a combination of chemical, physical, and biological processes in the systems. The removal capacity of a specific pollutant can be improved by altering design, operation schemes, and physical features of the system. TWs have proven relatively effective in removing nitrogen, organic matter, and suspended solids [8,9]. However, until now, sustained and reliable P removal has been difficult, as P has proven elusive [10–13]. The available P removal strategies in TWs include; sorption, chemical precipitation, sedimentation, and biomass uptake with subsequent harvest [14,15]. Nevertheless, these strategies have been evaluated with variable success, and it has been stated that the P removal is mainly determined by the sorption capacity of the material used in the main bed [13].

A filter installed outside the main bed (an external filter) containing a highly P sorbent material could overcome some of the P removal issues. The external filter would polish effluent from the TWs by removing the excess P, but also allow a straightforward material exchange once the filter material has become P saturated, in comparison to exchanging the entire main bed of the TW. Another advantage of the external filter is its ability to make P more accessible for reuse and research has indeed already shown the potential use of P-enriched filter materials as a slow-release P fertilizer [16,17]. The recovery of P is important, as P is a limited and non-renewable resource [18,19], but also an essential plant nutrient with P deficiency negatively affecting plant growth [20]. P fertilization of agricultural fields is therefore crucial to secure a sustainable food production [21,22], and its demand is increasing as the world population is steadily growing.

A material's potential to sorb P can be determined by fitting sorption models to data from batch experiments. However, long-term experiments, allowing both ad- and absorption processes to occur, and setups with real wastewater, are needed to fully evaluate the materials potential. Such experiments would confirm if the batch determined P sorption capacity is indeed achievable. The column experiments described in this study have been conducted as a follow up on a short-term batch experiment for a selection of calcareous materials with maximum P sorption capacities ($Q_{max}$) ranging from 5.6 to 35.1 mg P $g^{-1}$ dry mass (DM) [23]. These materials were selected due to their high Ca content, because Ca is one of several parameters of importance for the P removal capacity [12,24]. One of the selected materials was modified with a Sol–Gel coating to improve its physical stability. A trade-off between preserving the sorption capacity of the non-coated material and the thickness of the coating was found, with the thinner coating preserving more of the capacity [23].

The first objective of this paper was to evaluate the P removal performance and physical stability of the filter materials under long-term operation. The second objective was to estimate the total P sorption capacity (ad- and absorption processes) of the filter materials. The third objective was to evaluate if the total sorption capacity was in agreement with the maximum sorption capacity determined with isotherm experiments. Finally, we wanted to evaluate if Sol–Gel coating compromised the P removal ability of the materials.

## 2. Materials and Methods

### 2.1. Filter Materials and Their P adsorption Capacities

Eight materials were selected for the column experiment; six were supplied commercially and two were Sol–Gel coated materials (Table 1). Calcite (CAL) and Phosclean (PHO) have been developed specifically for P removal, while a natural product composed of silica-

calcite gravel, known as Opoka (OPO), has shown potential for P removal. Catsan (CAT) and Hygiene (HYG) are two commercially available cat litters, both calcium silicate gravel, composed of natural chalk and fine quality quartz sand. The last commercial material was quartz sand (SAN), which was included as a low Ca content material. These six materials will be referred to as non-coated materials. A coating process was performed to improve the physical stability of two materials, since similar materials have caused operational problems, e.g., disintegration during prolonged operation [25,26]. The two coated materials included in this experiment have been modified with an inorganic silica-based hydrosol coating by soaking the CAT material in a solution composed of the synthesized base coating diluted with demineralized water. The materials were exposed to the coating solutions for 5 min, but differed in the dilution ratios of 1:10 and 1:1 and will be denoted CAT A and CAT C, respectively. We refer to Jensen [23] for the full description of the coating procedure, which also determined the potential P sorption for the materials included in the column studies by fitting sorption models for short-term isotherm experiments. The $Q_{max}$ was determined by fitting a non-linear Langmuir model as:

$$q = (Q_{max} \times a \times C_{eq})/(1 + a \times C_{eq})) \tag{1}$$

with $q$ (mg P g$^{-1}$ DM) as the P removal of the material, with $C_{eq}$ as the equilibrium P concentration in the solution after 24 h, and with $a$ (L mg$^{-1}$) as a constant describing the materials P adsorption affinity (Table 1). Of the tested materials, CAT was the most promising material with the largest $Q_{max}$. However, when coating the CAT material, a trade-off between preserving the sorption capacity and the thickness of the coating was found. It was found that CAT materials with a thin coating preserved more of the sorption capacity compared to CAT materials with a dense coating.

**Table 1.** Material information and their maximum P adsorption capacities ($Q_{max}$). The materials include six non-modified materials and two coated materials, CAT A and CAT C, with CAT A having a thin layer Sol–Gel coating and CAT C having a denser coating layer. Material in column shows the average amount of material installed in the columns ($n = 2$, except CAL, $n = 1$). $Q_{max}$ were determined by fitting a non-linear Langmuir model in Jensen [23].

| Materials | Abbreviation | Company/Location, Country | Material in Column (kg) | $Q_{max}$ (mg P g$^{-1}$ DM) | General Description |
|---|---|---|---|---|---|
| Catsan | CAT | CATSAN®, UK | 0.82 | 35.1 | Commercial cat litter product. Calcium silicate gravel composed of natural chalk and fine quality quartz sand. |
| Catsan coating A | CAT A | Modified CATSAN, DK | 0.77 | 22.1 | |
| Catsan coating C | CAT C | Modified CATSAN, DK | 0.91 | 9.6 | |
| Hygiene | HYG | MULTIFIT®, DE | 0.62 | 30.2 | |
| Calcite | CAL | IMERYS Industrial Minerals, DK | 1.41 | 34.2 | Developed for P removal. Calcium carbonate granulates. |
| Opoka | OPO | Bełchatów, PL | 1.72 | 10.5 | Natural product. Carbonate silicate gravel produced from natural reserves. |
| Phosclean | PHO | NUWEN®, FR | 1.64 | 5.6 | Developed for P removal. Apatite granulates produced from a Moroccan natural product. |
| Quartz sand | SAN | Franzefoss a/s, DK | 2.54 | 0.7 | Natural sand extracted from a gravel pit in Denmark. |

## 2.2. Chemical and Physical Characterization

The element contents of each material were analyzed as described in Section 2.7 Element content. The initial Ca concentrations of each material ranged from 133 to 333 mg g$^{-1}$ DM (except for SAN) and the P concentration from 0.1 to 116 mg g$^{-1}$ DM. Variations in the concentration of Mg, Al, and Fe were also found, with Mg ranging from 0.7 to 16.8 mg g$^{-1}$ DM, Al ranging from 0.9 to 16.9 mg g$^{-1}$ DM, and Fe ranging from 0.6 to 10 mg g$^{-1}$ DM. The specific element concentrations for each material have been reported in Jensen [23], which also reported the materials' physical characterizations. In brief, HYG was the most porous material with the highest bulk porosity of 67% and the lowest specific weight of 0.35 g mL$^{-1}$, while SAN was the least porous material. Generally, only minor variations in the particle size distribution were observed, i.e., the effective size ranged from 0.21 to 0.68 mm and the uniformity coefficient (all with values of less than three) indicated that all materials had a uniform particle size distribution.

## 2.3. Experimental Column Setup

The experimental setup consisted of 15 individual columns (PVC sewage pipes; h = 22 cm, d = 10.5 cm) installed in a darkened laboratory at approximately 20 °C. The eight selected materials were installed in the columns (*n* = 2, except for CAL, *n* = 1) (Table 1; Figure 1). The columns were each filled with approximately 1600 mL of material (corresponding to 0.62–2.54 kg) and up-flow fed with tap water spiked with $KH_2PO_4$ (Table 1). The columns were fed in a continuous flow mode using a multi-channel feeding pump (RP-150 series, LACHAT instruments, Milwaukee, WI, USA).

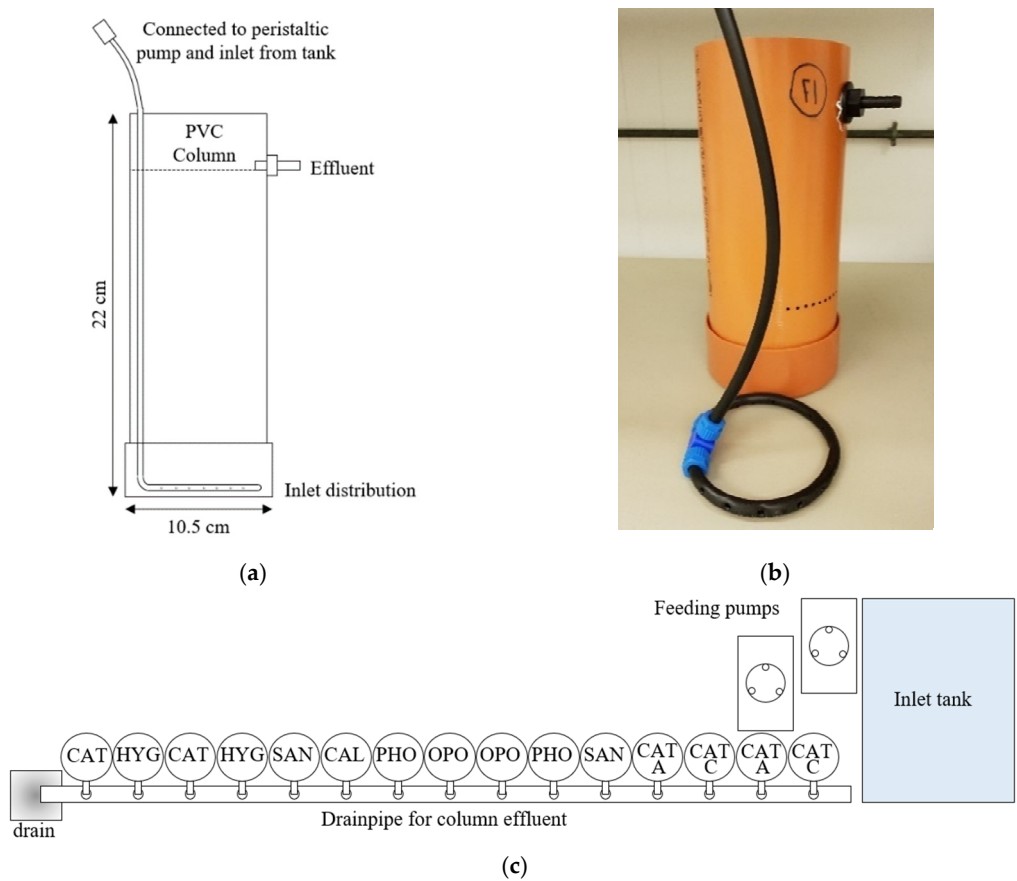

**Figure 1.** Column setup: (**a**) schematic overview of a single column set-up, (**b**) illustration of columns with inlet hose and inlet distribution, and (**c**) overview of the column setup.

### 2.4. Operational Periods

The experiment ran for 111 weeks in four different periods with subsequent increased hydraulic loading rate and inlet P concentration. Period I ran for a total of 52 weeks at a hydraulic loading rate of 18 mm d$^{-1}$ and a P concentration of 3.3 mg P L$^{-1}$ to ensure a stable performance of the materials (Table 2). Due to a development phase of the coated materials, the four columns with CAT A and CAT C were initiated 22 weeks into operation period I. These columns were fed at a higher hydraulic rate in the remaining weeks of period I to compensate for the late startup. Hereafter, period II was initiated and ran for 34 weeks with a P concentration of 5.5 mg P L$^{-1}$ and rate of 33 mm d$^{-1}$, and consecutively period III ran for 8 weeks with a rate of 204 mm d$^{-1}$. Finally, period IV ran 17 weeks with a P concentration of 21.5 mg P L$^{-1}$. The P solution was prepared weekly from a stock solution and tap water mixed directly in the inlet tank, which was supplied with a water pump to ensure proper mixing. The setup was checked several times a week and the distribution hoses were renewed twice a year to ensure reliable flow rates and operation.

**Table 2.** Characteristics of the P-spiked tap water for the four operative periods.

| Parameter | Operative Periods | | | |
|---|---|---|---|---|
| | I | II | III | IV |
| Weeks of operation | 52 | 34 | 8 | 17 |
| Accumulated weeks of operation | 52 | 86 | 94 | 111 |
| Surface hydraulic loading rate (mm d$^{-1}$) | 18 | 33 | 204 | 227 |
| Hydraulic retention time (d) | 3.8 | 2.8 | 0.4 | 0.4 |
| pH [a] | 8.3 | 8.8 | 8.1 | 7.6 |
| Conductivity [a] (mS cm$^{-1}$) | 0.56 | 0.52 | 0.56 | 0.53 |
| Temperature (°C) | 19.5 | 19.3 | 20.0 | 19.6 |
| Inlet P concentration (mg P L$^{-1}$) | 3.3 | 5.5 | 5.5 | 21.5 |

[a] Tap water had a pH of 8.5 and a conductivity of 0.50 mS cm$^{-1}$.

### 2.5. Sample Collection and Analysis

During the initial two periods, all effluent from the individual columns was collected in 1 L bottles and the amount measured one to three times a week. However, during period III and IV, the effluents from the columns were lead directed to a drain because of the high flow rates. A weekly sample of the effluent was collected in a bottle for four hours. In both cases, the discharge was weighed and used to calculate the flow rate (L d$^{-1}$) based on the period of sample collection. The surface hydraulic loading rate ($HLR_S$; mm d$^{-1}$) was calculated;

$$HLR_S = \frac{Q}{A} \tag{2}$$

with $Q$ as the flow rate (mm$^3$ d$^{-1}$) and with $A$ as the columns surface area (mm$^2$). The hydraulic retention time ($HRT$; d) was calculated as:

$$HRT = \frac{V \times \phi}{Q} \tag{3}$$

with $V$ as the volume of the column (mm$^3$), with $\phi$ as the individual bulk porosity (%) determined for the materials, and with $Q$ as the flow rate (mm$^3$ d$^{-1}$).

After collection, the pH and conductivity of the effluents were measured directly in the outlet bottle and an aliquot was extracted, filtered (5–8 μm, Qualitative Filter Paper, Frisenette, Knebel, Denmark) and after an adequate dilution analyzed for P concentration using the Molybdenum Blue Method [27]. The measured data for the P-spiked inlet tap water for the operative periods are summarized in Table 2.

### 2.6. P removal and P load

The P removal (mg P kg$^{-1}$ DM) by the materials were calculated individually for each column on a weekly basis from the difference in P concentration measured from the inlet tank water, the P concentration in column effluent, the flow rate, and the DM of the filter materials. This also allowed for calculation of the total P mass loading (mg P g$^{-1}$ DM), total P mass removal (mg P g$^{-1}$ DM), and the percentage of P removal of inlet concentration (%).

### 2.7. Element Content

After termination of the P loading, the water was drained from the columns and the columns left at room temperature (20 °C) until the material had dried (7 days). The columns were then divided into four height fractions, i.e., the top fraction (15–20 cm), the upper-mid fraction (10–15 cm), the lower-mid fraction (5–10 cm), and bottom fraction (0–5 cm). Starting from the top, every fraction was transferred to a pre-weighted foil tray, weighted, oven-dried to a constant weight (60 °C for 2 days), and reweighed to determine the mass (g DM) of the material. To ensure a representative and homogenized sample from the four fractions, the material was mixed in a tray and three subsamples (15 mL) were pooled in one sample (45 mL). The pooled sample for each fraction was ground in a MKM6003 rotating blade grinder (Bosch, Gerlingen-Schillerhöhe, Germany). Subsamples of the ground material (0.1 g DM) were digested in 4 mL nitric acid (65% $HNO_3$) and 2 mL hydrogen peroxide (30% $H_2O_2$) in a Multiwave 3000 microwave digestion system (Anton Paar, Graz, A). Blanks and reference material (tomato) samples were included in all analyses. The content of phosphorus (P), calcium (Ca), magnesium (Mg), iron (Fe), and aluminum (Al) were analyzed in an Optima 2000 DV ICP-OES (Perkin Elmer Instruments Inc., Shelton, CT, USA). A delta element content was calculated by subtracting the initial measured element content (in the original material) from the element content measured in the fractions after terminating the experiment. Hence, positive values indicate a higher element content while negative values indicate a lower element content.

### 2.8. Statistics

Graphs were prepared in SigmaPlot 12.5 (Systat Software Inc., San Jose, CA, USA) and in JMP 14 (SAS Institute, Marlow, UK).

## 3. Results

### 3.1. P removal during the Operation Period

Initially, all columns removed P effectively, as the P concentration measured in the column effluent was low (Figure 2). However, after seven weeks of operation, an increase in outlet P concentration was detected for SAN with effluent concentrations above 1 mg P L$^{-1}$. The increase in effluent P concentration continued during the following weeks and reached 14.3 mg P L$^{-1}$ after 111 weeks (Figure 2; Table 2). Likewise, after 80–90 weeks of operation, the columns with OPO and PHO also showed consistent increases in outlet P concentrations (Figure 2). The increase in effluent P concentration for PHO and OPO continued until the end of the experiment and reached effluent P concentrations of 1.3–1.5 mg L$^{-1}$ (Table 2). Only minor increases in outlet P concentration were observed for CAL and HYG from week 80 to week 111 (Figure 2). The effluent reached a P concentration of 0.05 mg P L$^{-1}$ for HYG and OPO after two years of operation (Table 2). The remaining materials (CAT, CAT A, and CAT C) had very low effluent P concentrations in the complete operation period and showed consistent P removal, only reaching a P concentration of 0.01 mg P L$^{-1}$ after the two years of operation (Figure 2; Table 2). Unfortunately, we had to terminate the experiment after two years of operation (before the columns were completely P saturated) due to resource and time constrains. Hence, we did not reach the point measuring similar P concentrations in effluent and inlet, and we could therefore not determine the columns materials total P removal capacity.

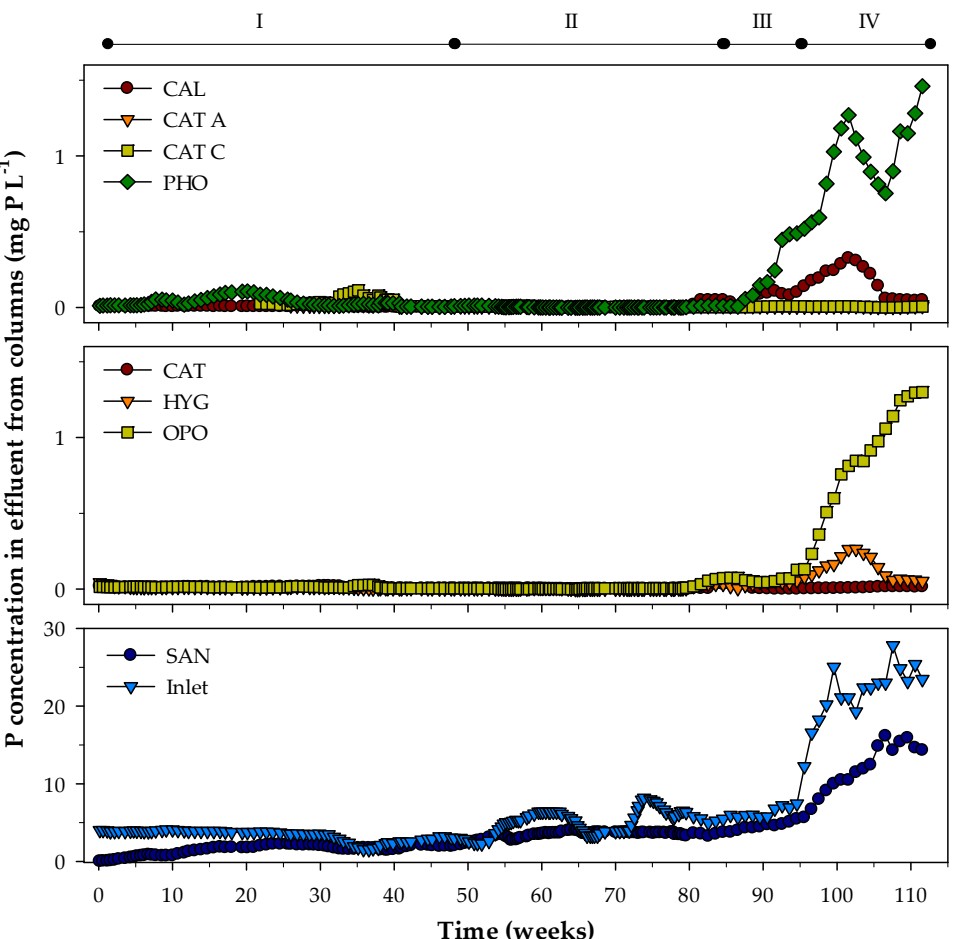

**Figure 2.** The P concentration in effluent from the 8 materials and in the inlet tank (running mean, *n* = 5). The operation period of I–IV is indicated at the top of the figure. Be aware of different y-axis scale for the SAN material and the inlet tank (lowest panel).

The P mass load and P mass removal varied for the materials depending on the amount of material installed in the columns (Table 3). HYG had the highest bulk porosity and therefore only 0.6 kg was installed resulting in a high specific P mass loading rate. Conversely, the lowest P mass loading rate was measured for SAN since these columns were filled with 2.5 kg material.

**Table 3.** Performance of the columns in the last week of operation. Values for P mass loading and P mass removal is an average (*n* = 2, except CAL, *n* = 1) ± Standard deviation.

| Material | Effluent P Concentration [a] (mg P L$^{-1}$) | P Mass Loading (mg P g$^{-1}$ DM) | P Mass Removal (mg P g$^{-1}$ DM) | P Removal (%) |
|---|---|---|---|---|
| CAL | 0.05 | 4.81 ± N/A | 4.78 ± N/A | 99.3 |
| CAT | 0.01 | 7.81 ± 0.26 | 7.80 ± 0.26 | 99.9 |
| CAT A | 0.01 | 7.95 ± 0.52 | 7.95 ± 0.52 | 100.0 |
| CAT C | 0.01 | 6.51 ± 0.77 | 6.51 ± 0.77 | 100.0 |
| HYG | 0.05 | 12.91 ± 3.39 | 12.84 ± 3.40 | 99.5 |
| OPO | 1.30 | 3.96 ± 0.24 | 3.33 ± 0.31 | 84.1 |
| PHO | 1.46 | 3.53 ± 0.13 | 3.39 ± 0.22 | 96.2 |
| SAN | 14.32 | 2.77 ± 0.19 | 1.12 ± 0.24 | 40.4 |

[a] Last value of running mean, hence, average of the three last measured values (*n* = 3). N/A = not applicable.

The P mass loading and the P mass removal were similar when comparing the materials that did not show or only showed minor increases in effluent P concentration (CAL, CAT, CAT A, and CAT C in Table 3). In contrast, some discrepancies between the P mass loading and the P mass removal were observed for the materials (HYG, OPO, PHO, and SAN) with increased P concentration in the effluent. This is in good agreement with the fact the materials did not show sustained P removal in the complete operation period, and, hence, some of the P loaded was not removed by the material (Figure 2; Table 3). A similar result was found in the P removal as percentage of the inlet concentration, which was especially low for SAN, but also showed removal < 100% for PHO and OPO (Table 3).

### 3.2. P content in the Column Fractions

The P content in the four height fractions revealed a clear pattern across all columns as the highest P contents were measured in the bottom fraction closest to the inlet distribution (Figure 3). In addition, the second highest P contents were measured in the consecutive mid fraction in all columns, i.e., the fraction just above the bottom fraction and second closest to the inlet distribution. Generally, the lowest contents were measured in the top fraction, while a more mixed pattern was apparent for the upper-mid fraction (Figure 3).

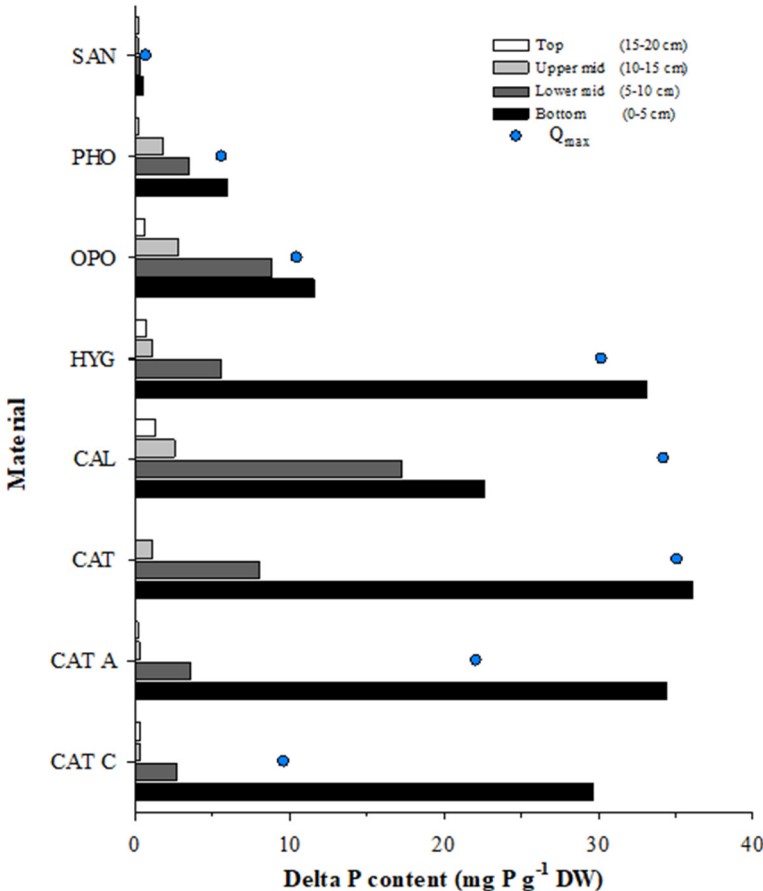

**Figure 3.** Delta P content in the four height fractions. The delta P content (mg P g$^{-1}$ DM) is the P content measured in the fractions after termination subtracted the initial measured P content (in the original material). A positive delta P indicates an accumulation of P while a negative delta P indicate a loss of P. The four fractions are: top (15–20 cm), upper-mid (10–15 cm), lower-mid (5–10 cm), and bottom (0–5 cm). Columns were up-flow fed. Bars show average ($n = 2$) except for CAL ($n = 1$). The maximum adsorption capacities ($Q_{max}$) were determined by fitting the non-linear Langmuir model in Jensen [23].

Figure 3 also shows the $Q_{max}$ measured by fitting a non-linear Langmuir model in a previous experiment. Several of the materials (CAT, OPO, PHO, and SAN) had P contents in the bottom fraction that matched well when compared with $Q_{max}$, hence, indicating a good agreement between the potential P sorption in the batch and the P sorption measured in the column experiment (Figure 3). CAT had a bottom fraction with a P content of 36.1 mg P g$^{-1}$ DM (the highest P content measured for all columns), which match well when compared with the measured $Q_{max}$ of 35.1 mg P g$^{-1}$ DM. Likewise, the bottom fraction P contents for OPO, PHO, and SAN were similar to the measured $Q_{max}$ values (Figure 3). More specifically, the bottom fraction contained 0.8, 5.9, and 11.6 mg P g$^{-1}$ DM for SAN, PHO, and OPO, respectively, while the $Q_{max}$ values were determined to 0.7, 5.6, and 10.5 mg P g$^{-1}$ DW, mentioned in the same order. In comparison, we observed a poor compliance for CAL between the $Q_{max}$ and the P content in the bottom fraction, as the $Q_{max}$ was measured to 34.2 mg P g$^{-1}$ DM and the bottom fraction only contained 22.6 mg P g$^{-1}$ DM. This indicated that $Q_{max}$ in the short-term experiment predicted too high a P sorption potential for CAL. The opposite was the case for HYG, CAT A, and CAT C, as more P accumulated in the bottom fraction compared to the reported $Q_{max}$ values. More specifically, the P content in the bottom fraction was measured to 29.7, 33.2, and 34.4 mg P g$^{-1}$ DM for CAT C, HYG, and CAT A, respectively, while $Q_{max}$ only were reported to 9.6, 30.2, and 22.1 mg P g$^{-1}$ DM (Figure 3). These results showed that for some materials there was not a direct connection between $Q_{max}$ determined in the batch experiments and the P sorption measured for the materials in the column experiment.

### 3.3. Remaining Elements in Column Fractions

The Ca content showed a height fractioned pattern with higher Ca contents measured at the bottom fraction closest to the inlet distribution system for all the columns, except for SAN and PHO (Figure 4). The pattern was especially evident for the five calcareous materials (HYG, CAL, CAT, CAT A, and CAT C), where an increase in Ca at the bottom of the columns, ranging from 50 to 80 mg Ca g$^{-1}$ DM were measured, when compared to the initial Ca content.

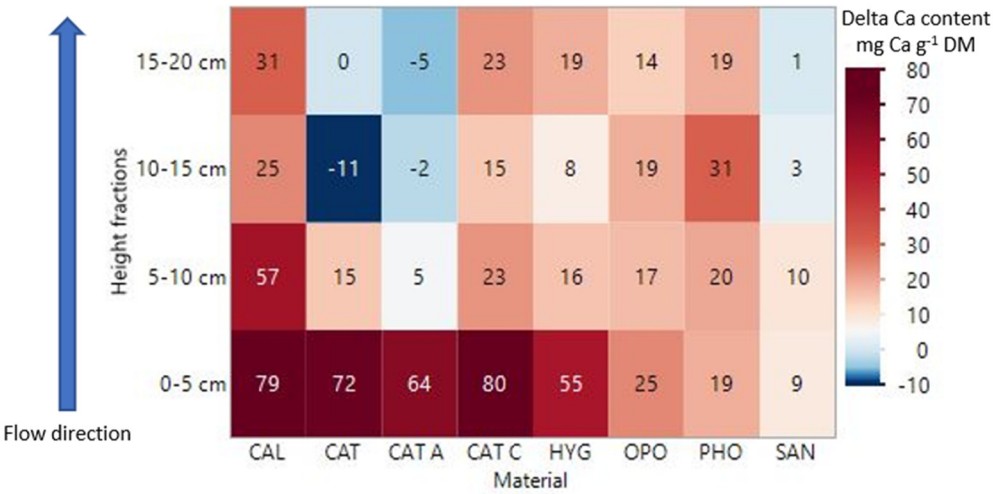

**Figure 4.** Delta Ca content in four height fractions. The delta Ca content (mg Ca g$^{-1}$ DM) is the Ca content measured in the fractions after termination subtracted the initial measured Ca content (in the original material). A positive delta Ca indicates an accumulation of Ca while a negative delta Ca indicate a loss of Ca. The four fractions are; top (15–20 cm), upper-mid (10–15 cm), lower-mid (5–10 cm), and bottom (0–5 cm). Columns were up-flow fed (indicated by blue arrow). Each square in the heatmap show an average (*n* = 2) except for CAL (*n* = 1).

Additionally, higher contents of P were measured in layers with higher Ca content, which was especially evident for the bottom fractions of CAL, CAT, CAT A, CAT C, and HYG (Figures 3 and 4). To assess this overlap, the relation between the delta P content

and the delta Ca content were plotted (Figure 5). The figure only shows the distribution for the bottom and the lower-mid fractions, as these are the layers of interest with both Ca and P accumulation. Furthermore, to elucidate to which of the calcium-phosphate precipitates this could have occurred, three calculated scenarios were included: (i) Apatite ($Ca_5(PO_4)_3OH$), (ii) Octacalcium phosphate ($Ca_4H(PO_4)_3$), and (iii) Dicalcium phosphate ($CaHPO_4$). In the scenarios, the proportion between the Ca to P molecular mass were calculated, giving (i) 4.4, (ii) 3.8, and (iii) 1.1, which was multiplied with the delta P content in the fractions to give the calculated delta Ca content. The distribution of the data in this study shows that some of the points with low delta P content and delta Ca content are located both above and below the scenario lines, but that most of the of points are located in-between the proposed scenario (ii) and (iii), with some points located closer to the one or the other scenario. This distribution suggests that a mixture of both calcium precipitates explains the coupled increases in the delta P and the delta Ca content.

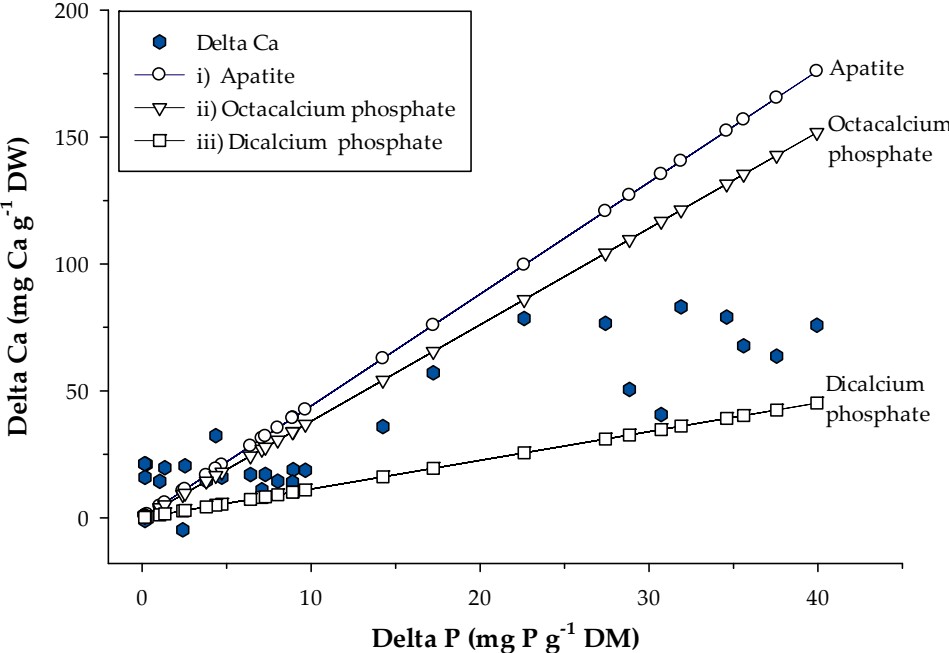

**Figure 5.** Delta Ca plotted against delta P. The blue filled symbols shows the delta Ca and the delta P content, which is the element content measured in the fractions after termination subtracted the initial measured element content (in the original material). The non-filled symbols shows three calculated delta Ca scenarios based on calcium-phosphate precipitation: (i) Apatite ($Ca_5(PO_4)_3OH$), (ii) Octacalcium phosphate ($Ca_4H(PO_4)_3$), and (iii) Dicalcium phosphate ($CaHPO_4$). See result text for more information about the scenarios.

A similar behavior, as for the Ca and P contents, were not registered for the Mg, Al, and Fe contents (results for Mg, Al, and Fe not shown). No major changes in the delta Mg content in any of the column fractions for CAL, HYG, OPO, PHO, and SAN were found, as only minor variations of $\pm$ 1.5 mg mg g$^{-1}$ DM were measured with no depth pattern. However, for the non-coated and coated CAT materials, higher Mg contents in the two mid-fractions were measured with a delta Mg content of 4 to 12 mg g$^{-1}$ DM. For HYG, CAT, CAT A, and CAT C, no changes in the delta Al content were measured. However, PHO generally had less Al content in all fractions ($-3$ mg Al g$^{-1}$ DM) and CAL, SAN, and OPO had slightly higher Al contents in all fractions in the range from 0.6 to 3 mg Al g$^{-1}$ DM with no distinct depth pattern. Moreover, no depth patterns were measured for Fe for CAT, CAT A, CAT C, HYG, and PHO as only minor deviations of $-0.1$ to 0.7 mg Fe g$^{-1}$ DM were measured. Generally, higher Fe contents of 1 to 3 mg Fe g$^{-1}$ DM were measured for CAL, OPO, and SAN in all fractions.

## 4. Discussion

### 4.1. The Calcareous Materials Showed Consistent P removal

The majority of the calcareous materials showed a consistent P removal under long-term operation. The CAT material consistently removed P effectively during the two years of operation, even at the high inlet P concentration of 21.5 mg P $L^{-1}$ and at retention times as short as 0.4 day. The coated materials, CAT A and CAT C, also showed consistent removal during the 90 weeks of operation. However, we observed a reduction in the P removal capacity for the other materials, as P was detected in the effluent after seven weeks for SAN as well as after 80–90 weeks for PHO, OPO, CAL, and HYG. SAN most notably showed the poorest P removal and, consequently, reached high P effluent concentrations. Nevertheless, none of the effluent P concentrations reached the same level as the inlet P concentrations, and the materials were therefore not completely P saturated, and determination of the materials' total P removal capacity was not possible.

### 4.2. All Materials Were Hydraulicly Stable in the Operation Period

All the materials were able to withstand the two years of operation, which indicated that the materials were hydraulicly stable and suitable for installation in external filters. However, we observed a negative delta Ca content in the upper fractions of CAT and CAT A, which could be explained by Ca leaching from the columns. This has been observed for calcareous materials, with initial Ca leaching of 270 mg $L^{-1}$ decreasing to 5 mg $L^{-1}$ in column effluents for Filtralite® P [28]. Additionally, the negative delta Ca for CAT A and CAT was not observed for CAT C, the material with the densest coating, and the delta Ca content was less negative for CAT A when compared with the non-coated CAT material. This difference in the delta Ca content for the CAT materials could be explained by the thickness of the coating, because a thinner coating led to a more negative delta Ca content, which was supported by the fact that the non-coated CAT material had the most negative delta Ca content. This could indicate that the Sol–Gel coating increased the stability of the materials. If the coating indeed minimized Ca leaching, the material's lifetime could be prolonged because of the improved mechanical stability. Furthermore, the minimized Ca leaching might also improve the material's P removal because the Ca content is related to the P sorption capacity of the materials. This highlights the importance of including Ca analyses of the inlet and effluent water samples to evaluate Ca leaching. However, further investigation of the physical stability is needed both during more realistic settings with columns fed with wastewater and during controlled measurements of the materials' stability.

### 4.3. P accumulated in the Bottom Fraction of the Columns

The delta P content for the fractions showed an accumulation in the bottom fraction with a decreasing delta P content towards the top. This P saturation pattern, with the highest P content closest to feeding point, is in agreement with other column experiments, such as the up-flow fed column experiment by Ádám [28] and the down-flow fed column experiment by Hylander [17]. The continuous decrease in P content in upper fractions indicated that only the bottom fractions were P saturated or close to P saturated. However, it can also be argued that at least two consecutive fractions have to obtain similar P contents before being able to conclude if the $Q_{max}$ of the fraction has indeed been reached [17].

An increase in the delta Ca content in the bottom fractions for most of the materials was also evident, and generally there was a coupled increase in the delta P and Ca contents. We argue that the combination between (i) the inlet tap water contributing with Ca and P, (ii) the leaching of Ca from materials, and (iii) the increased pH caused by the materials (as was observed in Jensen [23]) led to conditions favoring calcium-phosphate precipitates. As explored in the plot with the delta Ca and P contents (Figure 5), these precipitates could be a combination of several scenarios. The precipitates could explain some of the coupled increase detected in both delta Ca and P contents in bottom layers. However, we have not performed any investigation of calcium-phosphate precipitates in the columns and can

therefore not fully conclude on this matter. Further investigation should include not only analyses of calcium-phosphate precipitates in columns, but also in effluent water samples to evaluate if leaching of calcium-phosphate precipitations have occurred.

*4.4. Similarities and Discrepancies between P content in Bottom Fraction and $Q_{max}$*

A good agreement for the majority of materials were observed when comparing the P content in the bottom fraction with $Q_{max}$. Unfortunately, we did not reach the point of total saturation of the column materials, as inlet and effluent concentrations did not reach same P concentrations. Given that full P saturation of the columns was not achieved, we nevertheless argue that the bottom fractions for CAT, HYG, OPO, PHO, and SAN were P saturated because the P contents were similar to the measured $Q_{max}$. This suggests that the P sorption capacities determined in the short-term experiment were indeed comparable with the column studies, and the P retained in the bottom fraction confirms the P sorption capacity.

However, discrepancies between the P content in the bottom fraction and $Q_{max}$ for CAL were found, as the bottom fraction did not reach as high P content as the estimated $Q_{max}$. The opposite effect was found for CAT A and CAT C with the bottom fraction achieving higher P contents than estimated by $Q_{max}$. Discrepancies are not uncommon when comparing column and batch experiments. Drizo [29] performed long-term and short-term experiments for several materials and the study resulted in column experiment performing both worse and better in P sorption capacity when compared to the batch experiments. However, in this study, there was a predominance of similarities in P sorption capacities when comparing the $Q_{max}$ from batch experiments with the P content of the bottom fraction from column experiments, but as expected some discrepancies also occurred.

*4.5. Sol–Gel Coatings Had a Minimal Effect on the Materials P sorption Capacity*

The discrepancy between the P content in the bottom fraction and $Q_{max}$ for CAT A and CAT C showed that the Sol–Gel coatings had a minimal effect on the materials' $Q_{max}$. The P content in the bottom fractions was 1.6 and 3.1 times higher when compared to $Q_{max}$ for CAT A and CAT C, respectively. This shows that the two coated materials may achieve higher P sorption capacities than the short-term determined $Q_{max}$. Contrary to the findings in Jensen [23], where the thinner coating preserved more of the P sorption capacity. We therefore argue that the Sol–Gel coating had a minimal effect on the P sorption capacity, when the materials were installed under long-term operation.

*4.6. Applicable Aspects of the Filter Materials*

Assuming that P contents of the bottom fraction are comparable with the total P sorption capacity, the four most effective materials are HYG and non-/coated CAT with P sorption capacities in the range of 29.7–36.1 mg P g$^{-1}$ DM. Considering that domestic wastewater typically contains 1 kg P per Population Equivalents (PE) per year, the needed amount of material would be around 0.06–0.08 m$^3$ material PE$^{-1}$ yr$^{-1}$, based on the materials' specific weight and P sorption capacity. Under these assumptions and calculating using a lifespan of 10 years for an external filter, a single household (5 PE) would need an external filter of either (i) 2.9–3.6 m$^3$ if using non-coated/coated CAT or (ii) 4.2 m$^3$ if using HYG. External filters of these dimensions would correspond to an HRT of (i) 2 to 2.4 days if using non-coated/coated CAT or (ii) 3.4 days if using HYG, at a wastewater production of 60 m$^3$ PE$^{-1}$ yr$^{-1}$. These HRT rates fall in the ranges of HRT used in this study and seem realistic to obtain. Hence, we argue that the needed volumes of the calcareous materials are of an amount that is of a practical usage if an external filter is considered. Especially given that the bottom fractions of this column study confirm the potential P sorption capacity of both HYG and CAT, and also that the coated materials retained the high $Q_{max}$ found in the non-coated CAT material. On the contrary, OPO and PHO are not of any practical use, as we already measured values above 1 mg P L$^{-1}$ after only 80–90 weeks of operation, exceeding the P discharge limits. However, research looking into the performance of the

materials under more realistic settings for treating affected water is still needed to fully evaluate the P removal capacities. It can be expected that the filters most likely will saturate faster due to non-ideal hydraulics, or that other elements would co-sorb and lower the actual P sorbed by the materials. Another important aspect is that the filters would provide a simpler solution to the P removal limitations in nature-based solutions as when compared to the continuously dosing of coagulating chemicals as highlighted by Jenssen [30].

## 5. Conclusions

The calcareous materials showed a consistent P removal in the two years of operation, and all materials were hydraulicly stable throughout the period. The potential P sorption capacity was detected in most cases in the bottom layer when compared to $Q_{max}$ from the batch experiments, while in a few cases, the short-term experiment failed in estimating the materials' P sorption capacity. We further conclude that the Sol–Gel coating had a minimal effect on $Q_{max}$. This was explained partly because the P content in the bottom fractions for coated CAT materials were similar to the non-coated CAT, and partly because a trade-off between the coating thickness and preserving the P sorption capacity was not observed. Finally, we conclude that the calcareous materials used in this study constitute a high potential for use as external P filters for onsite wastewater treatment systems.

**Author Contributions:** Conceptualization, S.M.J., H.S., F.H.B., H.B. and C.A.A.; Methodology, S.M.J., H.S., F.H.B., H.B. and C.A.A.; Software, S.M.J. and H.B.; Validation, S.M.J., H.S. and F.H.B.; Formal Analysis, S.M.J. and H.B.; Investigation, S.M.J. and H.S. and F.H.B.; Resources, C.A.A. and H.B.; Data Curation, S.M.J.; Writing—Original Draft Preparation, S.M.J.; Writing—Review & Editing, S.M.J., H.S., F.H.B., H.B. and C.A.A.; Visualization, S.M.J.; Supervision, C.A.A. and H.B.; Project Administration, S.M.J. and C.A.A.; Funding Acquisition, C.A.A. and H.B. All authors have read and agreed to the published version of the manuscript.

**Funding:** The study was supported by the INCOVER EU project from the European Union's Horizon 2020 research and innovation program under grant agreement nº 689242 and INTEXT EU project funded from the European Union's LIFE18 ENV/ES/000233.

**Institutional Review Board Statement:** Not applicable.

**Informed Consent Statement:** Not applicable.

**Data Availability Statement:** The data presented in this study are available on request from the corresponding author. The data are not publicly available due to the ongoing work and planning of further development of the coated calcareous materials.

**Acknowledgments:** Solvei Mundbjerg Jensen acknowledges a Ph.D. scholarship from the Aarhus University Centre for Water Technology (WATEC) and Sino–Danish Centre for Education and Research (SDC).

**Conflicts of Interest:** The authors declare no conflict of interest. The funders had no role in the design of the study; in the collection, analyses, or interpretation of data; in the writing of the manuscript, or in the decision to publish the results.

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
