# Peer review of "Sustained Phosphorus Removal by Calcareous Materials in Long-Term (Two Years) Column Experiment"

_water, doi:10.3390/w14050682_

Round 1
Reviewer 1 Report
This paper reports the removal of P by Ca-based materials in a two-years leaching, which provides useful guidelines for water treatment. The experimental, results and discussions are generally good. Nevertheless, the paper would be more mechanistic if the adsorbed P-Ca can be characterized by spectroscopy, which may be addressed in future.
Author Response
Author’s response: Thank you very much for this feedback. We do agree that spectroscopy analyses could be interesting in future work, however, this technique was however not available to us.
Reviewer 2 Report
The manuscript entitled "Sustained phosphorus removal by calcareous materials in long-term (two years) column experiment" submitted to the journal Water deals with an important problem nowadays. The manuscript has been written clearly, and its content follows a similar work by the same authors, which was published in the journal Water at the end of last year.
There are several incorrect references to the cited literature that need to be corrected in the submitted manuscript.
Figure 4 is shown a total of three times (two must be removed).
Figure 2 may need to be modified. The range of the y-axis is inappropriately chosen, and due to the size of the displayed points, it is meaningless. I would consider merging into one figure (or into two - Inlet tank and SAN - separately) and would indicate each column in different color and symbol.
How do the authors explain the greater removal of P mass than the loading of P mass (at the same time, removal of P <= 100%) in Table 3? If the authors report a larger number of individual measurements (3), why do they not report a standard deviation?
Does the number of decimal places mentioned in Table 3 correspond to the precision of the determination? I cannot recalculate the removal of P in Table 3.
I have doubts about Figure 5. How do the authors explain the accumulation of calcium in the column when no calcium enters the column? Are the calculations provided by the authors correct? If calcium ions also entered the column (using non-distilled water), then this should also be determined or at least appropriately discussed. It would be good to discuss the composition of the individual column fractions not only in terms of the content of the individual elements but also in terms of the compounds present. Why not powder X-ray diffraction be measured, for example? This could provide at least a qualitative comparison of the columns before and after the accumulation of phosphorus. Powder X-ray could also clarify the form in which calcium and phosphorus are present.
Do the flows in the laboratory experiment correspond to the possible use in the domestic wastewater treatment plant? This aspect should also be discussed in conclusion, not just capacity.
Author Response
Author’s response: Thank you for your revision of the manuscript.
Reviewer 2.1: There are several incorrect references to the cited literature that need to be corrected in the submitted manuscript.
Author’s response: Thank you for calling it to our attention. These references to table 1 appeared when converting the word document to pdf and have now been corrected.
Reviewer 2.2: Figure 4 is shown a total of three times (two must be removed).
Author’s response: Again, this problem appeared when in the pdf conversion issues. It has been corrected in the revised version of the manuscript.
Reviewer 2.3: Figure 2 may need to be modified. The range of the y-axis is inappropriately chosen, and due to the size of the displayed points, it is meaningless. I would consider merging into one figure (or into two - Inlet tank and SAN - separately) and would indicate each column in different color and symbol.
Author’s response: If merging all the individual figures, a lot of the subtle differences in-between the materials will be lost. We have modified figure 2 into three figures, one with inlet and SAN, one with four materials and one with three materials. See line 238.
Reviewer 2.4: How do the authors explain the greater removal of P mass than the loading of P mass (at the same time, removal of P <= 100%) in Table 3? If the authors report a larger number of individual measurements (3), why do they not report a standard deviation?
Author’s response: Thank you for pointing out the differences. After carefully reconsulting the numbers, minor deviations in the calculations appeared. We have corrected and updated the numbers and added the standard deviation. The P mass removal and P mass loading is the accumulated number of the two years of operation, while the P removal (%) has been given for the last three sampling points.
Reviewer 2.5: Does the number of decimal places mentioned in Table 3 correspond to the precision of the determination? I cannot recalculate the removal of P in Table 3.
Author’s response: The removal of P (%) in table 3 was given as an average of the last three weeks (as indicated below the table), and this P removal is therefore different from the overall P removal (%) based on the P mass loading and P mass removal. We have chosen to use the overall P removal (%) and have updated table 3 according to this.
Reviewer 2.6: I have doubts about Figure 5. How do the authors explain the accumulation of calcium in the column when no calcium enters the column? Are the calculations provided by the authors correct? If calcium ions also entered the column (using non-distilled water), then this should also be determined or at least appropriately discussed. It would be good to discuss the composition of the individual column fractions not only in terms of the content of the individual elements but also in terms of the compounds present. Why not powder X-ray diffraction be measured, for example? This could provide at least a qualitative comparison of the columns before and after the accumulation of phosphorus. Powder X-ray could also clarify the form in which calcium and phosphorus are present.
Author’s response: Yes, the calculations are correct. As we already have included in the discussion in line 394, we argue that the tap water (non-distilled water) has introduced calcium ions to the columns, but leaching of Ca ions from materials could also explain some of the increase. We agree that powder X-ray diffraction could be an option to characterise the bound P on the media. However, unfortunately we did not have access to the X-ray diffractometer needed, and we have therefore not made any changes.
Reviewer 2.7: Do the flows in the laboratory experiment correspond to the possible use in the domestic wastewater treatment plant? This aspect should also be discussed in conclusion, not just capacity.
Author’s response: We have elaborated on flows and the hydraulic retention time in the discussion (line 438-441).
Reviewer 3 Report
The manuscript is well written and the research is designed properly. I have a few minor comments in the attached pdf file.

Author Response
Author’s response: Thank you very much this feedback, we highly appreciate your comments and have carefully addressed them in a point-by-point fashion below.
Reviewer 3.1: Avoid pointing in the abstract
Author’s response: The pointing in the abstract has been included based on layout of the journal of Water. Thus, no changes has been made.
Reviewer 3.2: maybe give this in months or days
Author’s response: Then numbers of weeks have been changed to months in the abstract.
Reviewer 3.3: wastewater.
Author’s response: water has been changed to wastewater.
Reviewer 3.4: check this reference (table 1).
Author’s response: The errors in references to table 1 have been corrected – apologies for the inconvenience.
Reviewer 3.5: In table 1, I would add here that e.g. OPOKA is a silica-calcite gravel and etc for other materials names.
Author’s response: Table 1 has been modified and the need information added.
Reviewer 3.6: (Table 2) Personally for me it is difficult to use weeks, I would rather put this numbers as days but as you prefer.
Author’s response: For table 2, the number of weeks have been kept.
Reviewer 3.7: Figure 4 reappearing.
Author’s response: The duplicates of figure 4 has been removed – apologies for the problems with reference system in word when converting to pdf.
Reviewer 3.8: Population equivalent.
Author’s response: Person equivalent has been changed to population equivalent.